# Relational Representation Learning

**Lucas Maes**[1,2*]**, Ian Hajra**[3]**, Arnesh Batra**[4]
**Hugues Van Assel**[5]**, Damien Scieur**[2,6]**, Randall Balestriero**[3]
[1]Mila, [2]Université de Montréal (DIRO),[3]Brown University,
[4]IIIT-Delhi, [5]Genentech, [6]Samsung SAIL

## Abstract

We introduce Relational Representation Learning (RRL), a unifying paradigm casting representation learning as a graph estimation problem. Instead of treating samples in isolation, RRL defines learning objectives through a relational graph encoding pairwise relationships between data points. An encoder learns by estimating this graph from embeddings and minimizing its discrepancy with a specified target graph. This perspective reveals that self-, semi-, and supervised learning can all be recovered as special cases of RRL, providing a single formalism that consolidates diverse pretraining objectives into a unified mathematical object. This view offers a principled lens for analyzing empirical observations in self-supervised learning, such as slow convergence, performance, and the projector–backbone accuracy gap. Our experiments show that increasing relational richness within the graph improves convergence speed and downstream performance, while clarifying the role of auxiliary projection heads. Code: https://github.com/rbalestr-lab/ssl-graphs

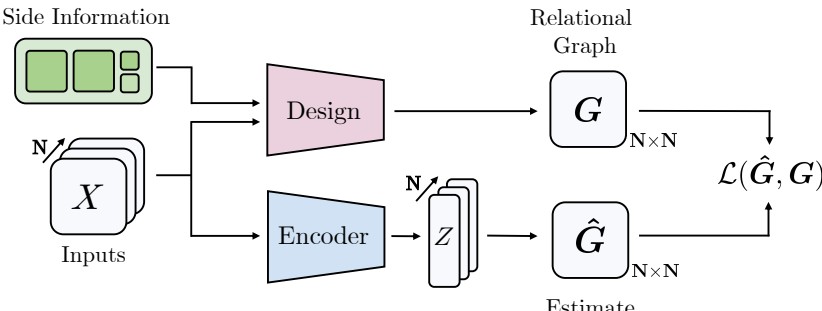

Figure 1: Relational Representation Learning (RRL). Given inputs $X$ and side information, a target graph $G$ is designed to specify pairwise sample relationships. The encoder learns by using its embeddings $Z$ to estimate a relation graph $\hat{G}$ and compares it to $G$ with the loss $\mathcal{L}$. Minimizing $\mathcal{L}(\hat{G}, G)$ encourages embeddings to capture the specified inter-sample relationships.

## 1   Introduction

Deep learning has historically focused on the dependency between samples $X$ and targets $Y$. In contrast, classical methods like kernel approaches (e.g., t-SNE [17]) or spectral embeddings (e.g.,

---

*Correspondence to `lucas.maes@mila.quebec`

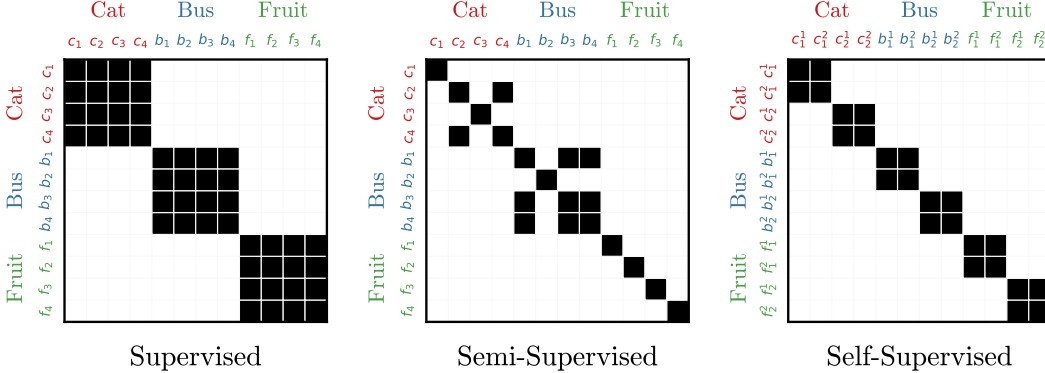

Figure 2: Relational graph associated with different representation learning paradigms. For simplicity, only binary pairwise relations are shown: black squares indicate a relation, white squares indicate none. (left) Supervised Learning target graph $G^{(\text{sup})}$ and (center) Semi-Supervised Learning partial-target graph $G_{\text{Semi}}$, relationships are determined based on shared class membership. Each graph uses 12 samples spread across 3 classes (cat, bus, fruit). (right) Self-Supervised Learning view graph $G_{\text{SSL}}$ only views originating from the same sample share a relationship.

ISOMAP [29], Laplacian Eigenmaps [5]) rely on similarity matrices rather than targets. These methods, however, require a meaningful input-space metric, which is often hard to define and computationally costly. Foundation models address these limitations by learning representations from massive datasets, capturing rich relational structures without handcrafted metrics. Training them has become central to modern AI, but each generation requires ad-hoc design choices, massive budgets, and opaque strategies for integrating modalities. Yet the diversity of approaches and rapid pace of development often hinders analysis and understanding. A unifying perspective on representation learning could support systematic comparison, clarify when current training tricks are effective, and advance AI research toward autonomous and reliable models [23].

**Main contributions.** We introduce **R**elational **R**epresentation **L**earning (**RRL**), a training paradigm unifying self-, semi-, and supervised learning that learns by predicting inter-sample relationships through their shared use of a relational graph rather than isolated features (fig. 1). **Section 2** formalizes RRL and provides examples of SSL objectives under relational graphs, while **Section 3** studies prominent Self-Supervised Learning (SSL) objectives in terms of losses over relational graphs, uncovering explicit links between graph quality and convergence speed, downstream performance, and the projector accuracy gap [6, 4].

## 2   RRL: Relationship Graphs Unify Representation Learning

We formalize RRL as a weighted graph $\mathcal{G} = (\mathcal{V}, \mathcal{E})$, with $\mathcal{V}$ being the set of nodes representing data samples $\boldsymbol{X} \triangleq [\boldsymbol{x}_1, \ldots, \boldsymbol{x}_N]^T \in \mathbb{R}^{N \times D}$ where $|\mathcal{V}| = N$, and $\mathcal{E}$ the set of weighted edges connecting samples based on the strength of their relationships. For the remainder of this manuscript, we consider a graph through its *symmetric* adjacency matrix $\boldsymbol{G} \in (\mathbb{R}^+)^{N \times N}$ with $(\boldsymbol{G})_{i,j} > 0$ iff samples $\boldsymbol{x}_i$ and $\boldsymbol{x}_j$ are known to share a relationship, $(i, j) \in \mathcal{E}$.

Building on this, our paradigm casts representation learning as a graph estimation problem, where the model $f_\theta : \mathbb{R}^D \mapsto \mathbb{R}^K$, commonly a Deep Network, learns by estimating the graph $\hat{\boldsymbol{G}}$ of relationships between samples from its embedding $\boldsymbol{Z} \triangleq [f_\theta(\boldsymbol{x}_1), \ldots, f_\theta(\boldsymbol{x}_N)]^T$, and compares it to a specified target relational graph $\boldsymbol{G}$, shown in fig. 1. The flexibility of our framework comes from **1)** the design of $\boldsymbol{G}$ and **2)** the loss function $\mathcal{L}(\boldsymbol{G}, \hat{\boldsymbol{G}})$, allowing it to incorporate many existing deep learning methods. By constructing the target relational graph from different sources, we can recover familiar paradigms (supervised, semi-supervised, and self-supervised), offering a unified perspective on representation learning through relational graphs.

## 2.1 A Unified View of Representation Learning

In supervised classification, the targets explicitly specify the shared relationship between a subset of samples, namely their class membership. Consequently, supervised learning can be viewed as RRL with the following target relational graph $G^{(\text{sup})} = YY^\top$, where $Y \in \mathbb{R}^{N \times C}$ is the one-hot target matrix for $C$ classes and $N$ samples, with $Y_{ij} = 1$ if sample $i$ belongs to class $j$ and $0$ otherwise. Semi-supervised learning is a special case of supervised learning, using only a subset of the data with available targets. It can thus be framed with the same target graph construction as $G^{(\text{sup})}$, but replacing $Y$ with a partial one-hot matrix $\tilde{Y}$ obtained from the available targets, where $\tilde{Y}_i = 0$ if sample $i$ has no associated target. We denote this semi-supervised partial-target graph as $G^{(\text{semi})} = \tilde{Y}\tilde{Y}^\top$. Unlike these approaches, self-supervised learning does not rely on human-annotated targets, but instead generates targets via various data augmentations. A sample $X_i$ is transformed into $V$ views (typically $V = 2$), all preserving the input semantics, yielding $\tilde{X} = [\tilde{x}_1^1, \ldots, \tilde{x}_1^V, \ldots, \tilde{x}_N^1, \ldots, \tilde{x}_N^V] \in \mathbb{R}^{NV \times D}$. The corresponding target graph is then defined as $G_{i,j}^{(\text{ssl})} = \mathbf{1}_{\{\lfloor i/V \rfloor = \lfloor j/V \rfloor\}} = I_N \otimes \mathbf{1}_{V \times V}$, where $i, j \in [NV]$, $\mathbf{1}_E$ denotes the indicator function of event $E$, $\otimes$ the Kronecker product, and $\mathbf{1}_{V \times V}$ a $V \times V$ matrix full of ones. In simple terms, $G^{(\text{ssl})}$ connects sample views originating from the same underlying sample, making it very sparse. All the graphs are illustrated in fig. 2.

## 2.2 Reframing SSL Objectives under Relational Graphs

To demonstrate the practical utility of this unified perspective, we now show how existing SSL methods can be recast within the RRL framework. We reformulate the objectives of SimCLR [9] and W-MSE [13] as functions of a relational graph $G$. We select these methods because (i) they are well-established and (ii) they represent the two dominant paradigms in self-supervised learning: contrastive and non-contrastive. Moreover, we provide additional graph formulations of established learning objectives in appendix D.

The relational formulation of the SimCLR objective given a target graph $G$ can be written as

$$\mathcal{L}_{\text{SimCLR}} = -\sum_{i=1}^{N} \sum_{j=1}^{N} (G)_{i,j} \log(\hat{G}(Z))_{i,j}, \quad \hat{G}(Z)_{i,j} \triangleq \frac{\exp(\tilde{z}_i^\top \tilde{z}_j)}{\sum_{k \in [N]} \exp(\tilde{z}_i^\top \tilde{z}_k)}, \quad \tilde{z} \triangleq \frac{z}{\|z\|_2}. \quad (1)$$

The relational formulation of the W-MSE objective given a target relational graph $G$ can be written as

$$\mathcal{L}_{\text{W-MSE}} = \sum_{i=1}^{N} \sum_{j=1}^{N} (G)_{i,j}\, \hat{z}_i^\top \hat{z}_j, \quad \text{where } \hat{z}_i \text{ are whitened and/or } \ell_2\text{-normalized features.} \quad (2)$$

## 3 RRL Explains Self-Supervised Learning Oddities

With our unified formulation, many self-supervised learning (SSL) phenomena, such as slow convergence, performance, and the projector–backbone accuracy gap, emerge naturally and can be explained in a principled way. In this section, we demonstrate how altering the structure of $G^{(\text{ssl})}$ provides a powerful tool for analysis. Using the relational formulations of SimCLR and W-MSE (section 2.2), we investigate: (i) the interplay between the relational richness of the target graph and properties such as convergence speed, downstream performance, and robustness; and (ii) the rationale behind the projector network, including the persistent accuracy gap between projector and backbone embeddings, an elusive phenomenon in the SSL literature. We provide implementation details in appendix B.

### 3.1 Interplay Between Graph Structure and SSL Properties

A major advantage of RRL is that it can analyze how implicit graph structure impacts SSL performance. The following sections analyze this impact across several key aspects. Additionally, we hypothesize the existence of a fundamental exploration/exploitation trade-off based on $G$ structure, shaping learning properties. We further discuss this tradeoff in appendix C.

**Convergence Speed & Performance.** SSL requires significantly longer training to converge compared to supervised learning, which we attribute to the sparsity of $G$ (fig. 3). Supervised learning relies on a coarse-grained graph defined by class membership, collapsing all samples in a class and allowing rapid convergence. In contrast, SSL performs instance discrimination on a fine-grained graph linking only augmented views of the same sample. This higher complexity slows convergence but encourages the model to learn robust, sample-specific features rather than coarse class semantics.

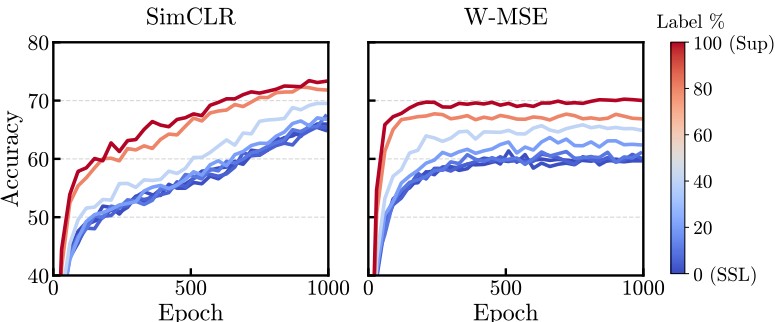

Figure 3: SSL linear probe accuracy of various relational graph richness. Reported results on CIFAR-100 for SimCLR and W-MSE using ResNet-18 as backbone. A richer relational graph increases performance.

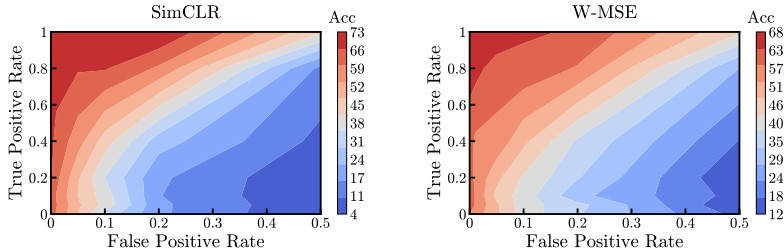

Figure 4: Sensitivity of relational graphs to corrupted edges on CIFAR-100 (ResNet-18). Contour plots show accuracy after 1000 epochs as a function of the True Positive Rate (TPR, the fraction of true edges recovered) and False Positive Rate (FPR, the fraction of false edges added) for SimCLR (left) and W-MSE (right). Accuracy is very sensitive to FPR: even a few incorrect links (e.g., "cat" to "truck") can harm representation quality.

**Sensitivity Analysis.** While deep learning is known for its robustness to target noise [26], it is unclear how the latter could hinder representation learning in RRL. We examine RRL's sensitivity to noise by introducing faulty relationships into $G$ during training and measuring performance degradation. We report our results in fig. 4. We find a pronounced noise sensitivity in sparse-relation regimes such as SSL, regardless of the algorithm. Contrastive learning method (SimCLR) can account for more noise than the Non-Contrastive method (W-MSE).

### 3.2 The Projector–Backbone Accuracy Gap

Current SSL methods use an auxiliary projector network atop the backbone, discarded after training. While the projector improves performance, it creates the so-called "projector-backbone accuracy gap" (fig. 5): backbone embeddings often outperform the projector on downstream tasks [14–16, 32]. We propose a complementary explanation: the projector compensates for the sparsity of $G$, enriching relational information during training. Empirically, the gap decreases as the number of relations increases, supporting this interpretation (see ablation in appendix F).

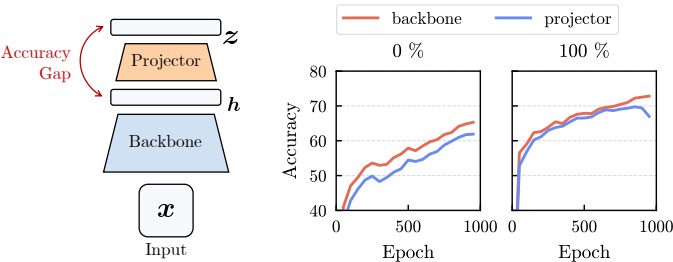

Figure 5: Projector–backbone accuracy gap. (Left) Backbone vs. projector embeddings. (Right) SimCLR projector accuracy gap on CIFAR-100 (1000 epochs, ResNet-18 backbone).

## 4 Conclusion

This work introduces Relational Representation Learning (RRL), a new paradigm that reframes representation learning as a graph matching task. This paradigm opens several promising directions. Future work could explore online estimation of relation graphs directly from embeddings, enabling adaptive and self-improving training. The graph formalism also suggests a natural path for multimodal representation learning, allowing seamless integration across modalities. Finally, theoretical analysis of relational graph properties may provide new guarantees and insights for large-scale models, helping improve reliability and understanding before deployment.

## Acknowledgments

LM and DS thank Prof. Simon Lacoste-Julien for his financial support, which funded LM during this project.

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

## A  Related work

**Unifying Representation Learning.**    Numerous studies have sought to unify the diverse approaches in representation learning. Within self-supervised learning (SSL), Garrido et al. [14] investigated the duality between contrastive and non-contrastive methods, showing that their performance differences can largely be attributed to parameter choices. More recently, Huh et al. [18] proposed a scaling-based perspective in which representational alignment emerges naturally with increasing model and dataset size. In parallel, Balestriero et al. [4] compiled a comprehensive overview of techniques and tricks that practitioners rely on across SSL methods. While valuable, these efforts remain largely confined to SSL-specific settings, leaving open the broader question of how to unify representation learning across domains.

**Inter-sample Graph.**    Building on these unifying perspectives, recent work has explored inter-sample relational structures to further enhance SSL. Several recent SSL approaches can be interpreted as performing implicit online graph estimation. For example, Dwibedi et al. [12], Koohpayegani et al. [20], Lebailly et al. [22] exploit nearest neighbors in the embedding space to replace augmentation views, thereby enriching the default $G^{(\text{ssl})}$ structure with additional relationships that improve training. A complementary line of work explicitly incorporates relational graphs into SSL. Sobal et al. [28] leverage LLM-generated pseudo-targets to construct inter-sample relations for SimCLR, showing promising gains despite the method's robustness to noisy edges. Cabannes et al. [7] formulate SSL as an active learning problem over graphs, studying how selective sampling affects representation quality. Finally, Balestriero and LeCun [3] unify classical SSL methods (SimCLR, VICReg, Barlow Twins) with spectral embedding algorithms, such as Laplacian Eigenmaps [5] and ISOMAP [29], highlighting the deep connection between self-supervised objectives and graph-based representation learning.

## B  Experimental Details

All models use a ResNet-18 backbone with a two-layer MLP projector. Our implementation builds on the `solo-learn` codebase [10], and we adopt the best hyperparameters from that work. We

trained for 400 epochs on CIFAR-10 and CIFAR-100 [21], and for 1000 epochs on ImageNet-100 [11]. All reported accuracies are validation accuracies, computed using an online linear probe on the embeddings.

## C    Discussion

**No Universal Relational Graph.**    There is no universal relational graph capable of solving every task as the optimal structure of such a graph depends entirely on the specific objective at hand. For example, addressing the CIFAR-10 classification problem requires collapsing samples from the same class together. Although effective for this task, this objective inherently produces representations that perform poorly on out-of-distribution datasets, with no way to reverse the collapse once learned. Consequently, a more "general" graph, i.e., one that imposes fewer constraints on the samples to be collapsed together, will tend to contain fewer edges, preserving greater flexibility in the learned representations. This is precisely the case for the relational graph that models self-supervised learning, which satisfies these looser constraints, and thereby produces features that generalize more effectively than those learned through supervised methods, albeit at the cost of slower convergence.

**Exploitation vs Exploration trade-off.**    Based on our observations, we conjecture the existence of a fundamental exploration–exploitation trade-off governed by the informational content encoded in the graph, measured, for instance, via a graph sparsity coefficient. A very sparse (or "general") graph imposes few constraints on which samples must be related, thus preserving flexibility for representation learning. In this regime, the model can rely more on its own inductive biases to discover useful features. In contrast, a dense (or "rich") graph, i.e., one that encodes many relationships, provides strong priors but limits the space of possible representations, pushing the model to exploit the information already present in the graph.

In this sense, *exploitation* corresponds to the use of inductive biases present in the data (and partially in the model) to solve the representation learning task, while *exploration* relies on the model's own inductive biases to uncover novel patterns not explicitly encoded in the graph. A metaphor for this distinction is the perception of flowers by a bee: equipped with ultraviolet vision, it can group flowers in ways that differ fundamentally from human classifications, revealing relationships invisible to us. More grounded in the AI research landscape, the shift from AlphaGo to AlphaZero reflects this trade-off. AlphaGo was primarily learned by exploiting inductive biases embedded in labeled expert human games. AlphaZero, by contrast, discarded human supervision entirely, exploring the strategy space through self-play and ultimately achieving superhuman performance.

**Practical Scalability of RRL.**    Kernel and spectral embedding methods face a well-known scalability limitation, as they require computing and storing an $N \times N$ matrix, where $N$ is the size of the dataset. At first glance, one might argue that our approach suffers from the same drawback, potentially limiting the practicality of RRL. In practice, however, it is sufficient to maintain only an $n \times n$ relational matrix, where $n$ is the mini-batch size. The resulting computational and memory overhead is negligible compared to the requirements of modern network architectures for forward and backward passes.

## D    More Relational Objectives

In this section, we derive the relational formulation of several more prominent objectives spanning Supervised, Semi-Supervised, and Self-Supervised Learning. A key advantage of RRL is its broad applicability, as it naturally accommodates both classical methods and recent advances, providing a unified mathematical foundation for diverse representation learning approaches.

### D.1    Supervised Learning

**SupCon.**    [19]
Assuming $\boldsymbol{G} \in \{0, 1\}^{N \times N}$ is hard target similarity matrix.

$$\mathcal{L}_{\text{SupCon}} = \mathcal{L}_{\text{SimCLR}} + \mathcal{L}_{\text{SuNCET}} \tag{3}$$

$$= -\sum_{i=1}^{N}\sum_{j=1}^{N}(\boldsymbol{G})_{i,j}\log(\hat{\boldsymbol{G}}(\boldsymbol{Z}))_{i,j} + \frac{1}{N}\log((\boldsymbol{G})_{i,:}\cdot\hat{\boldsymbol{G}}(\boldsymbol{Z})_{i,:}) \tag{4}$$

## D.2 Semi-Supervised Learning

**SuNCET.** [1]

Assuming $\boldsymbol{G} \in \{0,1\}^{N\times N}$ is hard target similarity matrix.

$$\mathcal{L}_{\text{SuNCET}} = -\frac{1}{N}\sum_{i=1}^{N}\log((\boldsymbol{G})_{i,:}\cdot\hat{\boldsymbol{G}}(\boldsymbol{Z})_{i,:}) \tag{5}$$

## D.3 Self-Supervised Learning

**DINOv2.** [25]

**Original DINO v2.** DINOv2 [25] formalized their ssl objective as the composition of three terms, two losses and one regularizer:

$$\mathcal{L}_{\text{DINOv2}} = \mathcal{L}_{\text{DINO}} + \lambda\mathcal{L}_{\text{iBOT}} + \lambda_2\mathcal{L}_{\text{koleo}} \tag{6}$$

where $\lambda, \lambda_2 \in \mathbb{R}$ are importance weight for secondary loss and regularizer term.

- **DINO Loss.** The original DINO loss term is a soft-clustering loss that promotes alignment between teacher and student cluster predictions. The cluster assigment distribution vector $\boldsymbol{p}_i$ is obtained after applying de-centering and/or softmax operations on the embedding $\boldsymbol{z}_i$.

$$\mathcal{L}_{\text{DINO}} = \frac{-1}{N}\sum_{i=1}^{N}\sum_{j=1}^{d}\boldsymbol{p}_{t,i,j}^{\text{cls}}\log(\boldsymbol{p}_{s,i,j}^{\text{cls}}) \tag{7}$$

- **iBOT Loss.** To make training more difficult. Some student patches are randomly masked during training. The iBOT loss [33] is a patch-level objective applied only to masked patches representation. It promotes similar soft-clustering between masked student patches representation and their corresponding representation from the teacher. iBOT loss aim to make the student model able to predict what should be present inside masked patches by predicting its cluster.

$$\mathcal{L}_{\text{iBOT}} = \frac{-1}{N}\sum_{i=1}^{N}\sum_{p=1}^{P}\sum_{j=1}^{d}\boldsymbol{p}_{t,i,j}^{p}\log(\boldsymbol{p}_{s,i,j}^{p})\mathbb{1}_{i,p} \tag{8}$$

with $P$ is the number of patches and the indicator function where $\mathbb{1}_{i,p} = 1$ if student sample $i$ has its patch $p$ masked.

- **KoLeo Regularizer.** Kozachenko-Leonenko (KoLeo) Regularizer [27] encourages normalized embeddings to spread uniformly by maximizing the log distance between each embeddings and its closest neighbor in the batch.

$$\mathcal{L}_{\text{koleo}} = \frac{-1}{N}\sum_{i=1}^{N}\log(d_i) \tag{9}$$

where $d_i = \min_{j\neq i}\|\boldsymbol{z}_{s,i}^{\text{cls}} - \boldsymbol{z}_{s,j}^{\text{cls}}\|_2$. Note that each $z_i$ is $\ell_2$ normalized.

**Graph DINO v2.** The original DINOv2 loss can be rewritten in a more general way by modeling sample relationships inside the batch itself. This can be done through a similarity graph $G \in \{0, 1\}^{N \times N}$ where $(G)_{ij} = 1$ if sample $i$ and $j$ are semantically related, e.g. through shared target.

$$\mathcal{L}_{\text{GraphDINOv2}} = \mathcal{L}_{\text{DINO}} + \lambda \mathcal{L}_{\text{iBOT}} + \lambda_2 \mathcal{L}_{\text{koleo}} \tag{10}$$

where

$$\mathcal{L}_{\text{DINO}} = \frac{-1}{N} \sum_{i=1}^{N} \sum_{j=1}^{N} \sum_{k=1}^{d} (G)_{ij} \boldsymbol{p}_{t,i,k}^{\text{cls}} \log(\boldsymbol{p}_{s,j,k}^{\text{cls}}) \tag{11}$$

$$\mathcal{L}_{\text{iBOT}} = \frac{-1}{N} \sum_{i=1}^{N} \sum_{p=1}^{P} \sum_{j=1}^{d} \boldsymbol{p}_{t,i,j}^{p} \log(\boldsymbol{p}_{s,i,j}^{p}) \mathbb{1}_{i,p} \tag{12}$$

with $P$ is the number of patches and the indicator function where $\mathbb{1}_{i,p} = 1$ if student sample $i$ has its patch $p$ masked.

$$\mathcal{L}_{\text{koleo}} = \frac{-1}{N} \sum_{i=1}^{N} \sum_{j=1}^{N} \log(d_{i,j}) \tag{13}$$

and

$$d_{i,j} = \min_{j \neq i, G_{ij}=0} \|\boldsymbol{z}_{s,i}^{\text{cls}} - \boldsymbol{z}_{s,j}^{\text{cls}}\|_2 \tag{14}$$

**SimDINOv2.** [31]

**Original SimDINO v2.** The original loss formulation of SimDINOv2 [31] is defined as

$$\mathcal{L}_{\text{SimDINOv2}} = \mathcal{L}_{\text{align}} + \lambda \mathcal{L}_{\text{rate}} \tag{15}$$

where $\mathcal{L}_{\text{align}} = \frac{1}{2} \|\boldsymbol{z}_1 - \boldsymbol{z}_2\|^2$ enforces similarity between positive examples and $\mathcal{L}_{\text{rate}}$ regularizes embeddings to be compact (i.e. low-dimensional) and information-efficient by removing noise redundancy. $\mathcal{L}_{\text{rate}}$, leverage the explicit coding rate regularization,

$$R_e(\boldsymbol{Z}) := \frac{1}{2} \log \det \left( \boldsymbol{I} + \frac{d}{\epsilon^2} \text{Cov}(\boldsymbol{Z}) \right), \tag{16}$$

**Graph SimDINO v2.** Assuming $N$ sample embeddings, each composed of $P$ patches of $d$ features, i.e. $Z \in \mathbb{R}^{N \times P \times d}$. The graph formulation of SimDINOv2 loss $\mathcal{L}_{\text{SimDINO}}$ can be written as:

$$\mathcal{L}_{\text{SimDINO}} = \mathcal{L}_{\text{align}} + \mathcal{L}_{\text{iBOT}} + \mathcal{L}_{\text{rate}} \tag{17}$$

with

$$\mathcal{L}_{\text{align}} = \frac{1}{N} \sum_{i=1}^{N} \sum_{j=1}^{N} (G)_{ij} \|\boldsymbol{Z}_{s,i,.}^{\text{cls}} - \boldsymbol{Z}_{t,j,.}^{\text{cls}}\|_2^2 \tag{18}$$

$$\mathcal{L}_{\text{iBOT}} = \frac{1}{N} \sum_{i=1}^{N} \sum_{p=1}^{P} \|\boldsymbol{Z}_{s,i,.}^{p} - \boldsymbol{Z}_{t,i,.}^{p}\|_2^2 \mathbb{1}_{i,p} \tag{19}$$

$$\mathcal{L}_{\text{rate}} = -\frac{\lambda}{2} \log \det \left( \boldsymbol{I} + \frac{d}{\epsilon^2} \text{Cov}(\boldsymbol{G}\boldsymbol{Z}_t^{\text{cls}}) \right) \tag{20}$$

where $\boldsymbol{Z}_s, \boldsymbol{Z}_t \in \mathbb{R}^{N \times P \times d}$ are the batched patch sequences embeddings from student and teacher branch respectively. Additionally, $\boldsymbol{Z}^{\texttt{cls}} \in \mathbb{R}^{N \times d}$ is the concatenation of all `cls` patches extracted from $\boldsymbol{Z}$, i.e. $\boldsymbol{Z}^{\texttt{cls}} = [\boldsymbol{z}_1^{\texttt{cls}}, \boldsymbol{z}_2^{\texttt{cls}}, \cdots, \boldsymbol{z}_N^{\texttt{cls}}]^\top$. Similarly, $\boldsymbol{Z}^p \in \mathbb{R}^{N \times d}$ is the concatenation of all patches $p$ extracted from batched sequences $\boldsymbol{Z} \in \mathbb{R}^{N \times P \times d}$, i.e. $\boldsymbol{Z}^p = [\boldsymbol{z}_1^p, \boldsymbol{z}_2^p, \cdots, \boldsymbol{z}_N^p]^\top$. Finally, $\mathbb{1}_{i,p}$ is the indicator function with $\mathbb{1}_{i,p} = 1$ if the patch $p$ from sample $i$ is masked in the student branch.

**NNCLR.** [12]

$$\ell_{i,j} = \frac{\exp(\text{sim}(\boldsymbol{q}(\boldsymbol{z}_i), \boldsymbol{z}_j)/\tau)}{\sum_{k=1, i \neq q} \exp(\text{sim}(\boldsymbol{q}(\boldsymbol{z}_i), \boldsymbol{z}_k)/\tau)} \tag{21}$$

$\boldsymbol{q}(\boldsymbol{z}_i) := \arg\min_{\boldsymbol{q}_i \in Q} \|\boldsymbol{q}_i - \boldsymbol{z}_i\|_2$

# E  Derivation of the Supervised Learning Relational Graph

In this section, we provide a theoretical justification for why the supervised learning relational graph takes the form $\boldsymbol{G}^{(\text{sup})} = \boldsymbol{Y}^\top \boldsymbol{Y}$, where $\boldsymbol{Y} \in \mathbb{R}^{N \times C}$ is the one-hot target matrix. This result emerges naturally from the solution to linear regression when cast as a projection operator.

**Proposition 1.** *Consider the supervised learning problem where we seek to predict targets $\boldsymbol{Y}$ from embeddings $\boldsymbol{Z}$. The optimal linear predictor in terms of mean squared error leads to a relational graph of the form $\boldsymbol{G}^{(\text{sup})} = \boldsymbol{Y}^\top \boldsymbol{Y}$.*

*Proof.* Consider the linear regression problem of predicting the one-hot target matrix $\boldsymbol{Y} \in \mathbb{R}^{N \times C}$ from embeddings $\boldsymbol{Z} \in \mathbb{R}^{N \times K}$:

$$\min_{\boldsymbol{W}} \|\boldsymbol{Y} - \boldsymbol{Z}\boldsymbol{W}\|_F^2 \tag{22}$$

where $\boldsymbol{W} \in \mathbb{R}^{K \times C}$ is the weight matrix and $\|\cdot\|_F$ denotes the Frobenius norm.

The optimal solution is given by:

$$\boldsymbol{W}^* = (\boldsymbol{Z}^\top \boldsymbol{Z})^{-1} \boldsymbol{Z}^\top \boldsymbol{Y} \tag{23}$$

The predictions are then:

$$\hat{\boldsymbol{Y}} = \boldsymbol{Z}\boldsymbol{W}^* = \boldsymbol{Z}(\boldsymbol{Z}^\top \boldsymbol{Z})^{-1} \boldsymbol{Z}^\top \boldsymbol{Y} = \boldsymbol{P}_{\boldsymbol{Z}}\boldsymbol{Y} \tag{24}$$

where $\boldsymbol{P}_{\boldsymbol{Z}} = \boldsymbol{Z}(\boldsymbol{Z}^\top \boldsymbol{Z})^{-1} \boldsymbol{Z}^\top$ is the projection matrix onto the column space of $\boldsymbol{Z}$.

To minimize the projection error, we want $\hat{\boldsymbol{Y}}$ to be as close as possible to $\boldsymbol{Y}$. The optimal case occurs when the embeddings $\boldsymbol{Z}$ perfectly capture the target structure, i.e., when $\hat{\boldsymbol{Y}} = \boldsymbol{Y}$. This requires:

$$\boldsymbol{P}_{\boldsymbol{Z}}\boldsymbol{Y} = \boldsymbol{Y} \tag{25}$$

Substituting the loss function with the projection formulation:

$$\mathcal{L} = \|\boldsymbol{Y} - \boldsymbol{P}_{\boldsymbol{Z}}\boldsymbol{Y}\|_F^2 \tag{26}$$

$$= \text{Tr}[(\boldsymbol{Y} - \boldsymbol{P}_{\boldsymbol{Z}}\boldsymbol{Y})^\top (\boldsymbol{Y} - \boldsymbol{P}_{\boldsymbol{Z}}\boldsymbol{Y})] \tag{27}$$

$$= \text{Tr}[\boldsymbol{Y}^\top \boldsymbol{Y}] - 2\text{Tr}[\boldsymbol{Y}^\top \boldsymbol{P}_{\boldsymbol{Z}}\boldsymbol{Y}] + \text{Tr}[\boldsymbol{Y}^\top \boldsymbol{P}_{\boldsymbol{Z}}^\top \boldsymbol{P}_{\boldsymbol{Z}}\boldsymbol{Y}] \tag{28}$$

Since $\boldsymbol{P}_{\boldsymbol{Z}}$ is a projection matrix, we have $\boldsymbol{P}_{\boldsymbol{Z}}^\top \boldsymbol{P}_{\boldsymbol{Z}} = \boldsymbol{P}_{\boldsymbol{Z}}$. Thus:

$$\mathcal{L} = \text{Tr}[\boldsymbol{Y}^\top \boldsymbol{Y}] - \text{Tr}[\boldsymbol{Y}^\top \boldsymbol{P}_{\boldsymbol{Z}}\boldsymbol{Y}] \tag{29}$$

$$= \text{Tr}[\boldsymbol{Y}^\top (\boldsymbol{I} - \boldsymbol{P}_{\boldsymbol{Z}})\boldsymbol{Y}] \tag{30}$$

To minimize this loss, we need to maximize $\text{Tr}[\boldsymbol{Y}^\top \boldsymbol{P}_{\boldsymbol{Z}}\boldsymbol{Y}]$, which encourages the projection $\boldsymbol{P}_{\boldsymbol{Z}}$ to preserve the structure encoded in $\boldsymbol{Y}^\top \boldsymbol{Y}$.

The matrix $\boldsymbol{Y}^\top \boldsymbol{Y}$ captures the target relationships: $(\boldsymbol{Y}^\top \boldsymbol{Y})_{ij} = \langle \boldsymbol{y}_i, \boldsymbol{y}_j \rangle = \mathbf{1}[\text{sample } i \text{ and } j \text{ have same target}]$, where $\boldsymbol{y}_i$ is the $i$-th row of $\boldsymbol{Y}$.

Therefore, by learning embeddings $\boldsymbol{Z}$ that minimize the projection error, we are implicitly learning to preserve the relational structure encoded in $\boldsymbol{G}^{(\text{sup})} = \boldsymbol{Y}^\top \boldsymbol{Y}$, which explains why this matrix naturally arises as the target relational graph for supervised learning. $\square$

## F  Projector Accuracy Gap

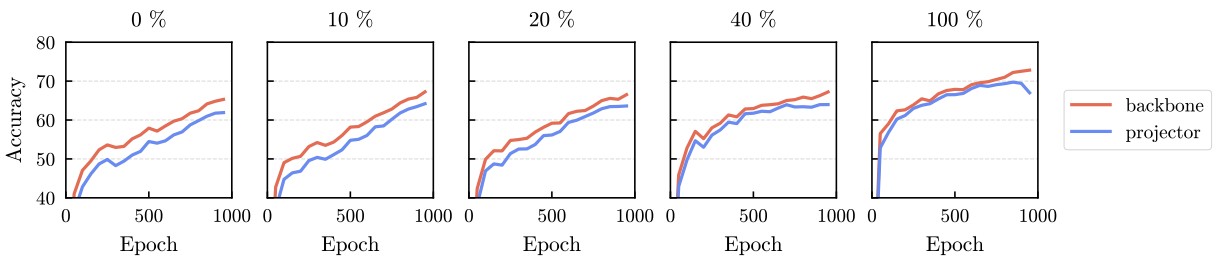

Figure 6: Projector accuracy gap for various percentages of target graph $\boldsymbol{G}^{(\text{sup})}$ for SimCLR on CIFAR-100. Increasing the number of relations in the target graph progressively reduces the gap.

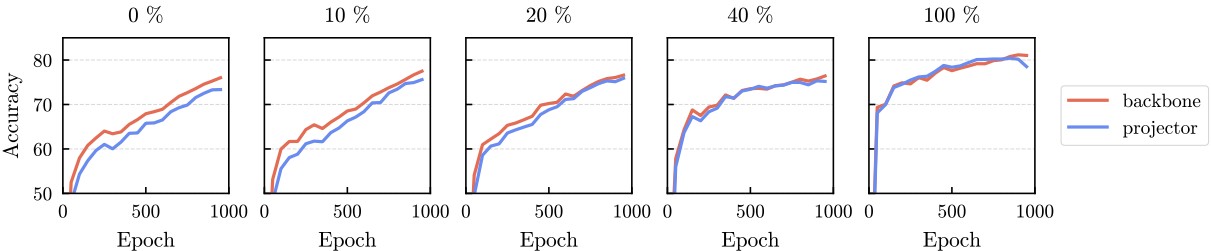

Figure 7: Projector accuracy gap for various percentages of target graph $\boldsymbol{G}^{(\text{sup})}$ for SimCLR on CIFAR-100. The target graph is constructed from the coarse target set (10-class). Increasing the number of relations in the target graph progressively reduces the gap.

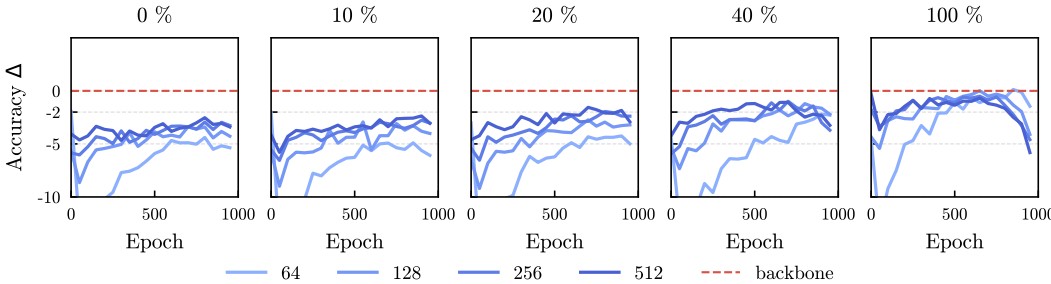

Figure 8: Projector-Backbone accuracy delta for various batch sizes and graph percentages of target graph $\boldsymbol{G}^{(\text{sup})}$ for SimCLR on CIFAR-100. Increasing the batch size as well as the number of relations in the target graph progressively reduces the gap.

## G  Online Graph Estimation

RRL accommodates a wide range of strategies for designing the target relational graph $\boldsymbol{G}$. A promising approach resides in learning to design $\boldsymbol{G}$. Recent works, like [28], took a first step in

that direction by leveraging Large Language Models (LLM) as an oracle to design $G$ by estimating similarities between data. One could also imagine designing $G$ entirely from the model's own representations and using it to guide its learning, a process known as self-distillation. This approach offers a powerful mechanism, allowing the model to bootstrap its performance by leveraging its own knowledge. Bootstrapping has already shown promising results in self-supervised learning (SSL) with methods such as NNCLR, MeanShift, and AdaSim [22], which utilize embedding nearest neighbors to refine learned representations. These approaches can be viewed as an implicit target relational graph design method, where the model relies on its own embeddings to infer pairwise relations and uses them to reinforce its representation learning. Yet, estimating the graph structure introduces the risk of erroneous pairwise relation predictions. A false positive, such as inferring a link between a "cat" and a "truck", can substantially impair representation learning. Section 3.1 unveils a pronounced sensitivity; an open question is therefore: *Under which condition can we use RRL for self-distillation?*

[22] observed the occurrence of representational collapse in self-distillation methods when incorporating nearest neighbor bootstrapping (NNB). This phenomenon is readily explained: NNB replaces one of the augmented images by retrieving the best neighbors from a queue of model-produced embeddings, which are initially of low quality. Early in training, NNB increases the likelihood of retrieving semantically unrelated samples, introducing harmful false positives, resulting in collapse. As training progresses, embeddings become more semantically aligned, reducing errors, improving NNB quality. This directly supports the adaptive strategy proposed by Lebailly et al. [22] based on latent space quality estimation. This effect arises naturally from the improved quality of the target relational graph, which in turn strengthens the training signal.

## H  Limitations of the Paradigm

Throughout this work, we highlighted the generality of the approach and its remarkable ability to recover many objectives across diverse areas of machine learning, from unsupervised to weakly and fully supervised settings. However, the graph-based approach is not exhaustive and does not capture certain important aspects of the studied methods. This is demonstrated in [2], which showed equivalences between self-supervised learning losses and spectral embedding methods, highlighting some key differences between the given closed forms and the representations obtained via SSL training.

First, even though the sample graph indicates which samples are related and which are augmentations of the same sample, it does not convey the core characteristics of the data augmentation, namely which features are perturbed and what invariances are enforced. A recent study [24] highlighted the great impact of choosing the correct augmentation in self-supervised learning. From a theoretical perspective, [30] showed that the data augmentation needs to be sufficiently aligned with the irrelevant features in the input data to discard them, with the alignment requirement depending on the family of method considered: joint embedding vs reconstruction.

Second, our formalism does not model the inductive bias of the network, which is known to play a key role in the obtained latent representation [8].

