# OpenReview forum: "Relational Representation Learning"
_NeurIPS.cc/2025/Workshop/UniReps — UniReps2025_

### Official Review · Reviewer_UTar · 2025-09-15
**Relational Representation Learning**

**Confidence:** 3

**Review:**

This work introduces the notion of Relational Representation Learning (RRL), which can be used to explain various existing techniques and open up several new directions.

I like the motivation and the general direction of the paper; the approach they are presenting sounds but the evaluation part of the paper is quite weak.

I will be glad to see the paper accepted to the conference, but adding more use cases and examples can make it much stronger

**Score:**

3

**Topic Fit:**

3

---

### Official Review · Reviewer_ZNND · 2025-09-15

**Confidence:** 4

**Review:**

## Summary

This work introduces “Relational Representation Learning” (RRL), a unifying framework that casts representation learning as graph estimation. A target “relational graph” encodes pairwise sample relations (labels, partial labels, or augmentations), and an encoder is trained to make its induced similarity graph the “relational graph.” This framing shows that supervised, semi-supervised, and self-supervised learning are just special cases of one paradigm. The paper also reformulates SimCLR and W-MSE as relational losses, and uses this lens to shed light on familiar SSL phenomena such as slow convergence (sparse graphs), sensitivity to noise (false edges), and the projector–backbone gap (projector compensates for sparse graphs). Experiments on CIFAR-100 support these claims, showing that richer graphs improve convergence and reduce the gap.

## Strengths

* **Unifying perspective**: The RRL perspective cleanly links supervised, semi-supervised, and SSL in one framework. Cross-paradigm unification is very valuable.
* **Explanatory value**: The framework offers some principled interpretations of SSL oddities (slow convergence, projector gap).

## Weaknesses / Limitations

* **Naming**: The naming “relational representation learning” collides with that of an already existing community. See, e.g., [this NeurIPS 2018 workshop of the same name](https://r2learning.github.io/).
* **Explanatory depth**: The account of the projector–backbone gap (as a response to sparse graphs) is appealing but remains somewhat high-level, and does not yet engage with other explanations (e.g., guillotine regularization, whitening effects).
* Scope of evidence: CIFAR‑100/ResNet‑18; even brief results/discussion on larger data or other modalities would help.
* What’s omitted by the formalism: Augmentation specifics and model inductive biases are not captured (the paper acknowledges this; App. H).

## Suggestions

* **Clarify the framing and naming**. Future versions could sharpen the scope of “relational”: emphasize the graph estimation viewpoint, and avoid confusion with existing uses of “relational learning” in other communities.
* **Strengthen the projector explanation**: A future version could probe the mechanism more directly. For example, varying graph sparsity while keeping architecture fixed, or comparing against known projector analyses (e.g. guillotine regularization). This would ground the relational explanation more firmly. Strengthen the claim that “denser GGG shrinks the gap” with a short controlled density sweep across datasets/batch sizes (your App. F–G trends are promising).
* **Generalization**. Experiments on more/larger datasets would help demonstrate the explanatory lens.

## References

[1] Bordes et al. Guillotine Regularization: Why removing layers is needed to improve generalization in Self-Supervised Learning. TMLR 2023.

**Score:**

3

**Topic Fit:**

3

---

### Official Review · Reviewer_qYba · 2025-09-16
**An interesting unifying framework that casts representation learning as a graph estimation problem, offering insights into SSL dynamics, but it faces significant limitations in originality and scalability**

**Confidence:** 4

**Review:**

## Strength

**S1. Conceptual clarity and unification**
- The paper introduces a straightforward and intuitive paradigm that unifies supervised, semi-supervised, and self-supervised learning within a single graph-based framework. This formulation effectively bridges modern deep learning methods with classical graph-based and spectral embedding approaches—an area of central importance in the self-supervised learning literature.
- Moreover, the work provides a solid and rigorous mathematical reformulation of existing objectives under the relational graph perspective, offering a clear and coherent formalism that is valuable.

**S2. Empirical and analytical contributions**
- The empirical analyses reveal consistent and interpretable trends, such as the observation that denser relational graphs promote faster convergence and improved downstream performance. The explanation of the projector–backbone gap through the lens of graph sparsity, supported by systematic ablations on relational richness and noise sensitivity, provides particularly useful insights for understanding and improving self-supervised learning.

## Weakness

**W1. Overlap with prior work**
- Several prior studies have already established unifying perspectives that are closely related to the proposed framework. For example, [1] demonstrated that both contrastive and non-contrastive self-supervised learning methods can be rigorously framed as classical spectral embedding techniques, providing closed-form solutions, insights into the role of pairwise relations, and a theoretical bridge between the two paradigms. Similarly, [2] introduced the explicit reformulation of SSL objectives in terms of a similarity graph and incorporates external oracles to inject relationships. Importantly, their work explicitly discusses extending the same formulation to supervised and semi-supervised regimes “depending on the oracle,” which bears strong resemblance to the proposed approach of “designing $G$ and then learning.”

**W2. Impracticality of handling an $n \times n$ graph**
- A central limitation of the proposed formulation is the reliance on constructing and operating over an explicit $n \times n$ relational graph. Such graphs scale quadratically in both memory and computation, quickly becoming infeasible for modern large-scale self-supervised learning. In practice, contrastive methods require a large and diverse set of negatives to avoid representational collapse. Restricting the graph to a batch-local $n \times n$ structure forces either prohibitively large batch sizes or the acceptance of low-quality negatives. For this reason, state-of-the-art methods have developed strategies to circumvent direct computation of the full similarity matrix, such as using memory banks [4], nearest-neighbor retrieval [3], or clustering-based contrastive assignments [5]. Establishing a clearer connection between the proposed method and these practical approaches would strengthen the paper’s claims of scalability and relevance to real-world SSL.

[References]
- [1] Balestriero, Randall, and Yann LeCun. *Contrastive and non-contrastive self-supervised learning recover global and local spectral embedding methods.* NeurIPS 2022.
- [2] Cabannes, Vivien, et al. *Active self-supervised learning: A few low-cost relationships are all you need.* ICCV 2023.
- [3] Dwibedi, Debidatta, et al. *With a little help from my friends: Nearest-neighbor contrastive learning of visual representations.* ICCV 2021.
- [4] He, Kaiming, et al. *Momentum contrast for unsupervised visual representation learning.* CVPR 2020.
- [5] Caron, Mathilde, et al. *Unsupervised learning of visual features by contrasting cluster assignments.* NeurIPS 2020.

**Score:**

2

**Topic Fit:**

2